

# Low serum albumin and total lymphocyte count as predictors of 30 day hospital readmission in patients 65 years of age or older

Robert Robinson

Department of Internal Medicine, Southern Illinois University School of Medicine, Springfield, IL, USA

## ABSTRACT

**Introduction.** Hospital readmission within 30 days of discharge is a target for health care cost savings through the medicare Value Based Purchasing initiative. Because of this focus, hospitals and health systems are investing considerable resources into the identification of patients at risk of hospital readmission and designing interventions to reduce the rate of hospital readmission. Malnutrition is a known risk factor for hospital readmission.

**Materials and Methods.** All medical patients 65 years of age or older discharged from Memorial Medical Center from January 1, 2012 to March 31, 2012 who had a determination of serum albumin level and total lymphocyte count on hospital admission were studied retrospectively. Admission serum albumin levels and total lymphocyte counts were used to classify the nutritional status of all patients in the study. Patients with a serum albumin less than 3.5 grams/dL and/or a TLC less than 1,500 cells per mm3 were classified as having protein energy malnutrition. The primary outcome investigated in this study was hospital readmission for any reason within 30 days of discharge.

**Results.** The study population included 1,683 hospital discharges with an average age of 79 years. The majority of the patients were female (55.9%) and had a DRG weight of 1.22 (0.68). 219 patients (13%) were readmitted within 30 days of hospital discharge. Protein energy malnutrition was common in this population. Low albumin was found in 973 (58%) patients and a low TLC was found in 1,152 (68%) patients. Low albumin and low TLC was found in 709 (42%) of patients. Kaplan–Meier analysis shows any laboratory evidence of PEM is a significant ($p < 0.001$) predictor of hospital readmission. Low serum albumin ($p < 0.001$) and TLC ($p = 0.018$) show similar trends. Cox proportional-hazards regression analysis showed low serum albumin (Hazard Ratio 3.27, 95% CI [2.30–4.63]) and higher DRG weight (Hazard Ratio 1.19, 95% CI [1.03–1.38]) to be significant independent predictors of hospital readmission within 30 days.

**Discussion.** This study investigated the relationship of PEM to the rate of hospital readmission within 30 days of discharge in patients 65 years of age or older. These results indicate that laboratory markers of PEM can identify patients at risk of hospital readmission within 30 days of discharge. This risk determination is simple and identifies a potentially modifiable risk factor for readmission: protein energy malnutrition.

Corresponding author
Robert Robinson,
rrobinson@siumed.edu

# INTRODUCTION

Hospital readmission within 30 days of discharge is a target for health care cost savings in the Medicare Value Based Purchasing (VBP) program through reductions in payments to hospitals with higher than expected readmission rates (*Centers for Medicare and Medicaid Services, 2015*). Because of the VBP initiative, health care organizations are investing considerable resources into efforts to reduce hospital readmission. Identifying patients at risk of hospital readmission can be accomplished with a variety of risk assessment tools that range from multidisciplinary patient interviews to simple screening tools using a handful of variables (*Kansagara et al., 2011*; *Silverstein et al., 2008*; *Smith et al., 2000*). These tools use risk factors such as age, ethnicity, socioeconomic status, severity of illness, previous hospitalizations and other factors to predict who is likely to be readmitted. Unfortunately, many risk factors for hospital readmission are not modifiable. Protein energy malnutrition (PEM) is a known and potentially modifiable risk factor for hospital readmission (*Lim et al., 2012*; *Fontes, Generoso & Toulson Davisson Correia, 2013*; *Sullivan, 1992*).

The prevalence of PEM in general medical inpatients is 20–45% depending on the method used to assess PEM (*Bistrian et al., 1976*; *Willard, Gilsdorf & Price, 1980*; *Persson et al., 2002*; *Barker, Gout & Crowe, 2011*). Similar rates of PEM can be found in surgical and intensive care patients (*O'Daly et al., 2010*; *Fontes, Generoso & Toulson Davisson Correia, 2014*; *Drevet et al., 2014*). The subset of patients with PEM have a greater likelihood of inpatient mortality (*Bonilla-Palomas et al., 2014*; *Persson et al., 2002*; *Barker, Gout & Crowe, 2011*; *O'Daly et al., 2010*; *Aziz et al., 2011*; *Correia & Waitzberg, 2003*; *Lim et al., 2012*), longer length of hospital stay (*Barker, Gout & Crowe, 2011*; *Jeejeebhoy et al., 2015*; *Aziz et al., 2011*; *Correia & Waitzberg, 2003*; *Lim et al., 2012*), hospital readmission (*Fontes, Generoso & Toulson Davisson Correia, 2014*; *Jeejeebhoy et al., 2015*; *Lim et al., 2012*; *Fontes, Generoso & Toulson Davisson Correia, 2013*; *Sullivan, 1992*; *Aziz et al., 2011*; *Friedmann et al., 1997*), and inpatient costs (*Correia & Waitzberg, 2003*; *Lim et al., 2012*). Several studies have shown that inpatient nutritional interventions can improve outcomes such as mortality (*Barchel et al., 2013*), length of stay (*Holyday et al., 2012*), and hospital readmission (*Holyday et al., 2012*).

Unfortunately, identifying patients with PEM is complex and there is no clear consensus method of identifying patients with PEM (*Kyle et al., 2006*; *Schneider & Hebuterne, 2000*). Pablo and colleagues conducted an analysis of four commonly used nutritional assessment tools and identified the Instant Nutritional Assessment (INA) as the most sensitive and specific tool for identifying people with malnutrition at the time of hospital admission (*Pablo, Izaga & Alday, 2003*). The INA has a sensitivity of 100% and a specificity of 92% in this study, and relies on only two commonly obtained laboratory tests: the serum albumin level and the total lymphocyte count (*Seltzer et al., 1981*).

The simplicity and objectivity of the INA as a nutrition screening tool is compelling. These laboratory tests are inexpensive and are likely to be performed on the majority of

patients ill enough to be admitted to a hospital. Several studies have shown good accuracy of the INA in predicting outcomes for femur fracture (*Symeonidis & Clark, 2006*; *O'Daly et al., 2010*). We hypothesize that the INA will also effectively identify medical patients at risk of hospital readmission within 30 days of discharge.

## MATERIALS AND METHODS

All medical patients 65 years of age or older discharged from Memorial Medical Center from January 1, 2012 to March 31, 2012 who had a determination of serum albumin level and total lymphocyte count on hospital admission were studied retrospectively. Memorial Medical Center is a 507 bed university-affiliated tertiary care center located in Springfield, Illinois, USA. Data on gender, age, serum albumin, DRG weight, total lymphocyte count (TLC), and hospital readmission within 30 days was extracted from the hospital electronic medical record system in a de-identified manner.

Admission serum albumin levels and total lymphocyte counts were used to classify the nutritional status of all patients in the study. Patients with a serum albumin less than 3.5 grams/dL and/or a TLC less than 1,500 cells per mm3 were classified as having PEM. Serum albumin and TLC were examined individually and together as predictors of outcome.

Diagnosis Related Group (DRG) weight was used as a marker of severity of illness. DRG weights are defined by the Centers for Medicare and Medicaid Services on an annual basis and are related to the cost and complexity of inpatient medical care for a specific DRG.

The primary outcome investigated in this study was hospital readmission for any reason within 30 days of discharge.

Institutional review board review for this study was obtained from the Springfield Committee for Research Involving Human Subjects. This study was determined to not meet criteria for research involving human subjects according to 45 CFR 46.101 and 45 CFR 46.102.

### Statistical analysis

Serum albumin and TLC were investigated as predictors of hospital readmission within 30 days. Qualitative variables were compared using Pearson $chi^2$ or Fisher's exact test and reported as frequency (%). Quantitative variables were compared using the non-parametric Mann–Whitney U or Kruskal–Wallis tests and reported as mean $\pm$ standard deviation. Rates of survival were evaluated by the Kaplan–Meier method and compared using the log-rank test. Demographic and clinical variables were included as explanatory variables in a Cox proportional-hazards regression analysis in the following manner:

1. Age, gender, DRG weight
2. Age, gender, DRG weight, low albumin
3. Age, gender, DRG weight, low TLC
4. Age, gender, DRG weight, low albumin, low TLC

Statistical analyses were performed using SPSS version 22 (SPSS Inc., Chicago, IL, USA). Two sided *P*-values <0.05 were considered significant.

**Table 1 Patient characteristics.**

|  | No PEM<br>N = 267 | PEM<br>N = 1,416 |  |
|---|---|---|---|
| Age in years (SD) | 77 (8.03) | 79 (8.42) | P < 0.001 |
| Female gender (%) | 161 (60%) | 780 (55%) | P = 0.115 |
| DRG weight (SD) | 1.00 (0.39) | 1.26 (0.72) | P < 0.001 |
| Serum albumin g/dL (SD) | 3.82 (0.23) | 3.12 (0.61) | P < 0.001 |
| Total lymphocyte count 1,000 cells/mm$^3$ (SD) | 3.50 (14.50) | 1.12 (1.01) | P < 0.001 |
| Readmitted within 30 days (%) | 13 (4.9%) | 206 (14.5%) | P < 0.001 |

**Table 2 Comparison of 30 day readmission-free survival between patients with normal and low albumin.**

| Group | Discharged | 10 days | 20 days | 30 days |
|---|---|---|---|---|
| Normal albumin | 709 | 700 | 682 | 669 |
| Low albumin | 972 | 871 | 824 | 793 |

**Table 3 Comparison of 30 day readmission-free survival between patients with and without PEM.**

| Group | Discharged | 10 days | 20 days | 30 days |
|---|---|---|---|---|
| No PEM | 266 | 261 | 258 | 253 |
| PEM | 1,415 | 1,304 | 1,248 | 1,209 |

## RESULTS

The study population included 1,683 hospital discharges with an average age of 79 years. The majority of the patients in this sample were female (55.9%) and had a DRG weight of 1.22 (0.68). In this sample, 219 patients (13%) were readmitted to the same hospital within 30 days of hospital discharge.

Protein energy malnutrition was common in this population. Low albumin was found in 973 (58%) patients and a low TLC was found in 1,152 (68%) patients. Low albumin and low TLC was found in 709 (42%) of patients. Patients with PEM were older, had a higher DRG weight, and were more likely to be readmitted to the hospital within 30 days than patients without evidence of PEM (Table 1).

Kaplan–Meier analysis shows any laboratory evidence of PEM is a significant ($p < 0.001$) predictor of hospital readmission (Fig. 1 and Table 2). Low serum albumin (Fig. 2 and Table 3, $p < 0.001$) and TLC (Fig. 3 and Table 4, $p = 0.018$) show similar trends.

Multiple Cox proportional-hazards regression models were constructed to investigate the relationship of low albumin and low TLC to hospital readmission (Table 5). The regression model had $C$-statistics ranging from 0.562 to 0.653. The model that included age, gender, DRG weight, low albumin, and low TLC had the highest $c$-statistic at 0.653. Further analysis of this Cox proportional-hazards regression model showed low serum

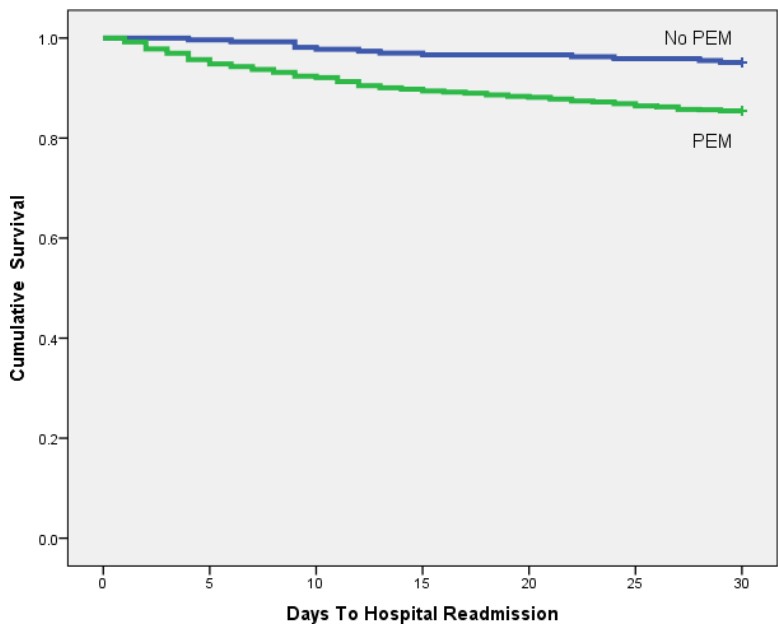

**Figure 1  Kaplan-Meier plot comparing 30 day readmission rates between patients with and without PEM.**

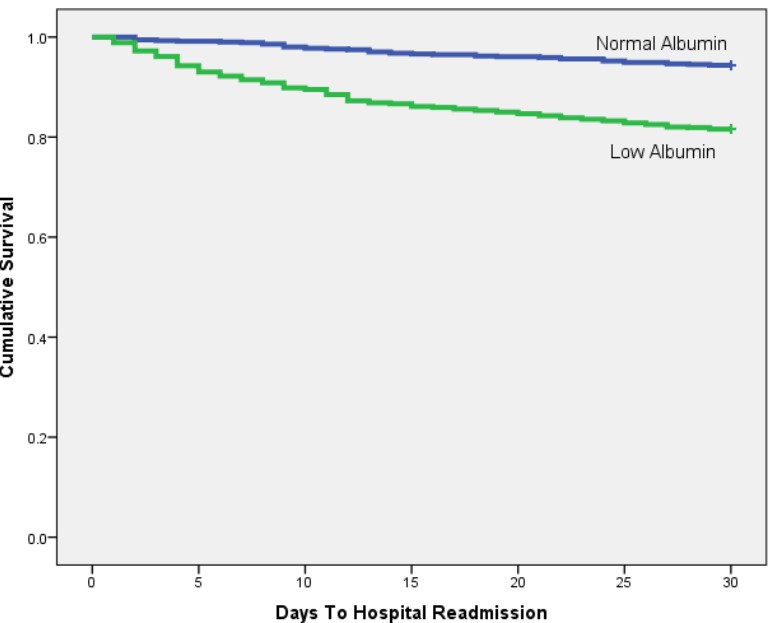

**Figure 2  Kaplan-Meier plot comparing 30 day readmission rates between patients with low and normal albumin levels.**

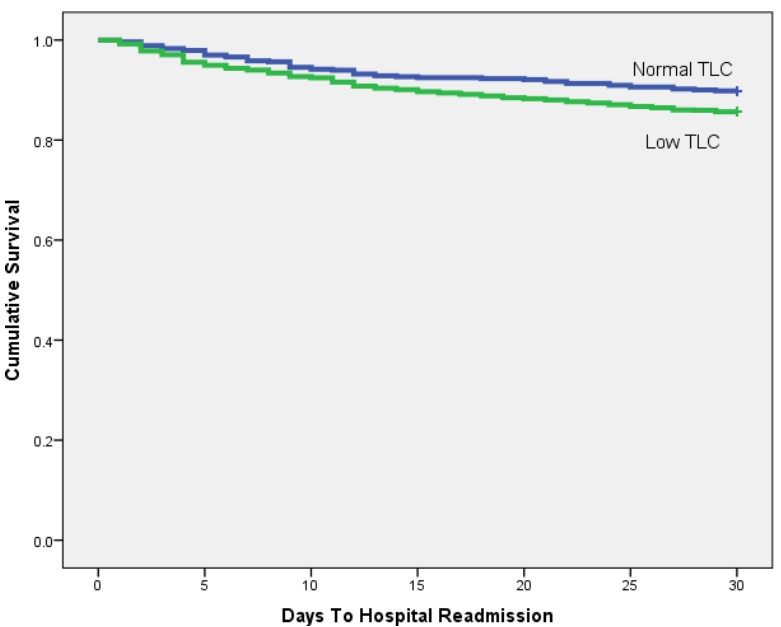

**Figure 3** Kaplan-Meier plot comparing 30 day readmission rates between patients with low and normal TLC.

**Table 4** Comparison of 30 day readmission-free survival between patients with normal and low TLC.

| Group | Discharged | 10 days | 20 days | 30 days |
|---|---|---|---|---|
| Normal TLC | 530 | 500 | 489 | 476 |
| Low TLC | 1,151 | 1,065 | 1,017 | 986 |

**Table 5** Cox proportional-hazard regression model characteristics for hospital readmission.

| Model components | −2 log likelihood | C-statistic |
|---|---|---|
| Age, gender, DRG weight | 3,209.726 | 0.562 |
| Age, gender, DRG weight, low albumin | 3,153.231 | 0.646 |
| Age, gender, DRG weight, low TLC | 3,204.440 | 0.570 |
| Age, gender, DRG weight, low albumin, low TLC | 3,150.560 | 0.653 |

albumin (Hazard Ratio 3.27, 95% CI [2.30–4.63]) and higher DRG weight (Hazard Ratio 1.19, 95% CI [1.03–1.38]) to be significant independent predictors of hospital readmission within 30 days (Table 6).

## DISCUSSION

This study investigated the relationship of PEM to the rate of hospital readmission within 30 days of discharge in patients 65 years of age or older. These results indicate that laboratory markers of PEM can identify patients at risk of hospital readmission within 30

**Table 6  Cox proportional-hazards regression analysis of risk factors for hospital readmission.**

| Variable | Regression coefficient | Standard error | Wald | P value | Hazard ratio (95% CI) |
|---|---|---|---|---|---|
| Age | −0.009 | 0.008 | 1.17 | 0.280 | 0.99 (0.98–1.01) |
| Gender | −0.026 | 0.137 | 0.04 | 0.849 | 0.97 (0.75–1.27) |
| DRG weight | 0.177 | 0.073 | 5.85 | 0.016 | 1.19 (1.03–1.38) |
| Low albumin | 1.184 | 0.178 | 44.16 | <0.001 | 3.27 (2.30–4.63) |
| Low TLC | 0.254 | 0.159 | 2.57 | 0.109 | 1.29 (0.95–1.76) |

days of discharge. This risk determination is simple and identifies a potentially modifiable risk factor for readmission: protein energy malnutrition.

The results of this study show that PEM identified via the INA, is an independent risk factor for hospital readmission within 30 days even when controlling for age, gender and severity of illness. The INA is moderately good at predicting hospital readmission ($c$-statistic 0.653) which is comparable to other published methods for assessing readmission risk in real time ($c$-statistics range from 0.53 to 0.61, *Kansagara et al., 2011*). Malnutrition may not be the only cause of low serum albumin in this patient population because many illnesses can alter serum albumin values (*Bachrach-Lindstrom et al., 2001*; *Fleck, 1988*). Low serum albumin level alone is predictive of longer hospital stays (*Herrmann et al., 1992*; *Numeroso, Barilli & Delsignore, 2008*; *Jellinge et al., 2014*), readmission (*Herrmann et al., 1992*) and mortality (*Herrmann et al., 1992*; *Lyons et al., 2010*; *Barchel et al., 2013*; *Jellinge et al., 2014*; *Cabrerizo et al., 2014*) in medical inpatients.

More complex measures of nutritional status such as the Subjective Global Assessment (SGA) or Mini-nutritional Assessment (MNA) incorporate patient history, physical exam findings, and assessments of functional status (*Dempsey & Mullen, 1987*). These more complex measures of PEM are used in many studies of the clinical impact of nutritional interventions. However, the cost and time required to administer the SGA and MNA limits their widespread use. The INA, which we used in this study, was shown to be superior to the SGA in a small comparative study investigating the prevalence of PEM in hospitalized adults (*Pablo, Izaga & Alday, 2003*) and is suitable for adaptation as an automated clinical decision support tools within an electronic health record system to identify patients at increased risk of hospital readmission.

The generalizability of this investigation is limited because of the retrospective single center nature of this study. Hospital readmission data was only available from the study hospital, which could miss patients who were admitted at another hospital within 30 days of discharge. This would lead to a falsely low rate of hospital readmission. Unaccounted for local care delivery variables may have a significant impact on hospital readmissions within 30 days.

## CONCLUSIONS

Protein energy malnutrition in medical inpatients is a significant predictor of hospital readmission within 30 days of discharge. Further investigation focusing on comorbid

conditions is required to better understand the utility of laboratory tests of PEM for predicting hospital outcomes.

### Funding

The author received no funding for this work.

### Competing Interests

The author declares there are no competing interests.

### Author Contributions

- Robert Robinson conceived and designed the experiments, performed the experiments, analyzed the data, contributed reagents/materials/analysis tools, wrote the paper, prepared figures and/or tables, reviewed drafts of the paper.

### Human Ethics

The following information was supplied relating to ethical approvals (i.e., approving body and any reference numbers):

Institutional review board approval for this study was obtained from the Springfield Committee for Research Involving Human Subjects. This study was classified as an exempt study because no individually identifiable information was collected.

### Supplemental Information

Supplemental information for this article can be found online at http://dx.doi.org/10.7717/peerj.1181#supplemental-information.

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
