# Peer review of "Low serum albumin and total lymphocyte count as predictors of 30 day hospital readmission in patients 65 years of age or older"

_PeerJ, doi:10.7717/peerj.1181_

## Round 0.1 · original submission · Major Revisions

Overall, there is agreement that this is a well designed study. However, the reviewers raised a number of important points and I believe that an effect to address these constructive suggestions will significantly enhance the final manuscript.

Reviewer 1 ·

Basic reporting

Protein energy malnutrition (PEM) is identified as a modifiable risk factor for readmission within 30 day. PEM can be assessed using different tools. This paper uses blood levels which are routinely checked upon admission.
It is known that PEM is a risk factor for readmission. It was also shown that the inpatient nutritional interventions improve outcome such as mortality. PEM can be assessed in many ways. Instant Nutritional Assessment (INA, based on total lymphocyte count and serum albumin like in this study) it was sown to predict the femur fracture. This study hypothesises that PEM defined based on total lymphocyte count (TLC) and albumin is a predictor for hospital readmission. Furthermore, since these factors are easily assessed and PEM is a modifiable factor this study could help in lowering the readmission rate.
The authors supply the dataset. The article is clearly written.

Experimental design

Design is appropriate.

Validity of the findings

The major concern is that TLC does not seem to impact the hospital readmission. On the other hand the albumin is a known risk factor for hospital readmission as the authors mention in the discussion (lines 152-155). If this is the case then what is the impact of this paper.

Additional comments

All the abbreviations need to be defined before using them, including the abstract. The DRG weight is not defined anywhere from what I could see. In the abstract it should be specifically mentioned that 1.22 is the mean of DRG weight and 0.68 is its standard deviation.
The sensitivity and specificity mentioned on lines 49-51 are based on which gold standard?
The analysis was performed using the time to event specific techniques. While this is not incorrect, since all patients have at least 30 days follow-up, the analysis can be performed using the binary outcome defined as readmission within 30 days (yes/no). The advantage of this approach is that one could assess the predictability of the model using the C-statistics. For time to event analysis, the assessment of predictability is not that straightforward. However, as a means for illustration the Kaplan-Meier curves could still be shown if the authors choose to do so.
The authors conclude that the albumin and TLC predict for hospital readmission. I am wondering if indeed there is a need for both of these factors or albumin may be sufficient. Thus, it would be useful to have C-statistics for the following models:
1) Age, Gender, DRG weight
2) Age, Gender, DRG weight and Albumin
3) Age, Gender, DRG weight and TLC
4) Age, Gender, DRG weight, Albumin and TLC
Along these lines, it is not clear whether the C-statistics presented in lines 99-103 and Table 3 are based on models adjusted for Age, Gender and DRG weight.
Minor comments
Line 73. Is this sentence correct? This study was determined to not meet criteria for ….
Line 151. The ‘and’ at the end of the line is not necessary.

Reviewer 2 ·

Basic reporting

No Comments

Experimental design

For the results reported in Table 3, the prediction models should be described. The evaluation methods (such as training and testing or cross-validation strategy) to compute the sensitivity and specificity should be reported.

Validity of the findings

In Figures 1,2 and 3, it is better to list the number of patients at each time point (Days To Hospital Readmission)

For the results reported in Table 1, what are the values in bracket. It looks the values in the bracket have different units, such as, 77(8.03) vs 161 (60%).

Additional comments

This study was well-designed. The results from the study have significantly clinical meaning.

---

## Round 0.2 · accepted · Accept

The author successfully addressed all the questions and issues. The modified manuscript is much improved.